# Recycling of Carbon Fiber-Reinforced Composites—Difficulties and Future Perspectives

**DOI:** 10.3390/ma14154191

**Published:** 2021-07-27

**Authors:** Dragana Borjan, Željko Knez, Maša Knez

**Affiliations:** 1Laboratory for Separation Processes and Product Design, Faculty of Chemistry and Chemical Engineering, University of Maribor, Smetanova ulica 17, 2000 Maribor, Slovenia; dragana.borjan@um.si (D.B.); zeljko.knez@um.si (Ž.K.); 2Laboratory for Chemistry, Faculty of Medicine, University of Maribor, Taborska ulica 8, 2000 Maribor, Slovenia

**Keywords:** composite materials, carbon fiber-reinforced composites, recycling techniques, chemical recycling, subcritical fluids, supercritical fluids, supercritical alcohols

## Abstract

Carbon fiber-reinforced composites present an exciting combination of properties and offer clear advantages that make them a perfect replacement for a spread of materials. Consequently, in recent years, their production has dramatically increased as well as the quantity of waste materials. As future legislations are likely to prevent the use of landfills and incineration to dispose of composite waste, alternative solutions such as recycling are considered as one of the urgent problems to be settled. This study presents the leading technologies for recycling carbon fiber-reinforced composites, focusing on chemical recycling using sub- and supercritical fluids. These new reaction media have been demonstrated to be more manageable and efficient in recovering clean fibers with good mechanical properties. The conventional technologies of carbon fibers recycling have also been reviewed and described with both advantages and drawbacks.

## 1. Introduction

As carbon fiber-reinforced composites present an exciting combination of properties such as corrosion resistance, durability, low thermal expansion, high strength-to-weight ratios, and strength, their demand has increased in many industrial fields over the past few decades, from architecture to infrastructure and automotive [1,2]. They offer clear advantages that make them a perfect replacement for a spread of materials, including aluminum, granite, steel, and wood. Consequently, composites are fast becoming the material of choice [3]. In 2019, the European composites market size was approximately USD 17.88 billion. Expansion at a compound growth rate of 7.5% per year from 2019 to 2025 is anticipated. Therefore, reaching USD 27.54 billion by 2025 is expected [4]. However, each sector does not manifest the same interest in composites [1]. To illustrate, in aerospace and aircraft, the selection of materials is motivated by their performance and fuel efficiency [5]. This makes the high stiffness and relatively low weight of carbon fibers a desirable replacement. Conversely, in general engineering and surface transportation, the employment of carbon fibers is determined by cost constraints, usually less critical performance need, and high production rate requirements.

The crucial drawback of composites is that they are challenging to recycle, especially carbon fiber-reinforced composites due to their hardness and chemical stability [6]. However, the significant commercial value of carbon fiber-reinforced composites recycling lies in recovering long high modulus fibers with a high intrinsic value for their reuse in high-grade applications. Turning carbon fiber-reinforced composites waste into a valuable resource and shutting the loop in their life cycle is vital for the continued use of the material in some applications [7,8].

All of this motivates new research and developments to enhance the recyclability of composite materials. Some review papers on composite recycling are available in the literature [9,10,11,12,13,14], but composite recycling remains a relatively new area. Currently, chemical, mechanical, and thermal methods have been used to recycle composites [15,16]. A chemical process is used to recover both the clean fibers and fillers and depolymerized matrix in monomers or petrochemical feedstock. Mechanical recycling is mainly based on crushing, grinding, milling, and shredding the composite part into smaller pieces, which can then be further ground into powder. Thermal recycling involves heat breaking the scrap composite and combusting the resin matrix, thereby recovering the carbon fibers. This paper seeks to examine the implementation of engineering optimization techniques in composite recycling focused on novel techniques that use sub- and supercritical fluids.

## 2. Difficulties with Composite Disposal

The utilization of composite materials has simplified modern life. Still, the extensive use in every industrial segment has caused a rapidly increasing amount of waste and high issues globally despite all its advantages. Additionally, most carbon fiber-reinforced composites produced thus far are still operational. Still, familiar sources of waste include end-of-life components, manufacturing cut-offs, out-of-date prepregs, production tools, as well as testing materials.

The conventional way to handle composite waste is incineration or disposal in landfills. However, new European waste directives on landfills and incineration will pressure these traditional disposal routes for composite materials. To preserve the environment, legislation must be instigated, usually combined with economic instruments, such as taxes, to enforce recycling [17].

Environmental, as well as economic, are the crucial aspects for recycling directions today [18]. Thermoplastics have a possible environmental concern since they are mainly non-biodegradable and challenging to obliterate naturally [19]. On the other hand, materials reinforced with virgin carbon fiber-reinforced composite are not likely to be used in many products because of the high cost. Carbon fibers represent an expensive raw material; their price as of 2020 is 5–20 USD/square meters [20]. These days, researchers concentrate on looking for new composite materials and developing better, optimized fabrication processes to lower the fabrication costs and upgrade the material properties [21,22]. However, excluding future issues with composite disposal, it is also economically favorable to implement composite recycling. Recycling activities supply the potentiality to use low-priced carbon fibers for applications that do not demand high strength and eventually open sustainable and secure sources of carbon fiber material. Concern for the environment, limiting the utilization of finite resources, and the need to manage waste disposal, have led to increasing pressure to recycle materials at the end of their useful lives [23]. To ease the continued use of the composite material in some applications, such as wind turbine blades and the automotive industry (e.g., the BMW Group uses recycled carbon fibers for the manufacturing of the reinforcement of the C-pillar with sheet molding compound, and Toyota uses Mitsubishi Rayon’s sheet molding compound material for the manufacturing of the hatch door frame), it is crucial to modify composite waste into a valuable resource and close the loop within the composite life cycle [7,24,25]. Certainly, the enormous use of composite materials, thanks to their outstanding characteristics, leads to a rise in the quantity of waste produced.

Even though there are many valuable utilizations for thermoset composite materials, recycling at the end of the life cycle could also be a more complex issue [26]. Nonetheless, the perceived insufficiency of recyclability is increasingly essential and seen as a critical barrier for developing or even using composite materials in some markets. In addition to their specific issues, there are other problems related to recycling any material from end-of-life components, such as the necessity to handle contamination and collecting, identifying, sorting, and separating the scrap material (Figure 1). Furthermore, in their publication, Pompidou et al. (2012) discussed the relationship between recyclers and customers linked by an exchange of carbon fibers at the end-of-life stage [2].

As a result of composites’ complex composition (fibers, matrix, and fillers), it is tough to fractionate them into elemental components. Composite waste is mainly disposed of in landfills or incinerated with none recycling approaches. Albeit composite waste is relatively inert in regard to other waste (they produce no leachate and methane gas), replacements should be taken to reduce the quantity of waste disposed of and thus bring down the effect on the environment.

## 3. Usual Recycling Procedures

To comply with the legislation, manufacturers of composite products should have products for real recycling solutions. Several methods have been tried and classified into three categories: mechanical grinding to thermal and chemical degradation of resins. Some of the most usual recycling procedures are shown in Figure 2 and described below.

### 3.1. Mechanical Recycling

Mechanical recycling involves the use of crushing, grinding, milling, and/or shredding techniques. All the constituents of the primary composite are minimized in size to particles with a length from 50 μm to 10 mm [27]. The resulting scrap pieces are mixtures of fiber, polymer, and filler and can be segregated by sieving into powdered products (rich in resin) and fibrous products (rich in fibers) [28,29].

Mechanical recycling can be used as a charge, or partial reinforcement in other products leads to limited incorporation in new materials [30]. It is suitable for scrap composite material that is relatively clean and uncontaminated and from the known origin. It presents few advantages as it recovers both fibers and resins without using or producing hazardous materials. Moreover, this technique is more suitable for glass fiber-reinforced composites. Even though the technologies developed give powder and fibrous recyclates, which can be reused, the powder recyclates have the meager potential for reuse back into the original thermoset compounds. The fibrous recyclates have some prospective reinforcement materials, but they are not as good as virgin reinforcement. Furthermore, there are problems related to the bonding of the recyclate with polymers, and usually, they do not fit the thermosetting polymers.

Finally, the degradation of mechanical properties of the recovered fibers is significant; their architectures are coarse, non-consistent, and unstructured, and hence, the possibilities for their re-manufacturing are limited [11].

### 3.2. Thermal Recycling

Thermal recycling processes involve heat to break the scrap composite down and combust the resin matrix, thereby recovering the carbon fibers. The most used thermal recycling techniques are pyrolysis and fluidized bed procedure.

A fluidized bed operates by balancing the downward gravity forces of the weight of the particles in the charge with the upward details created by the high gas flow. This process implies thermal decomposition of the polymer matrix followed by the release and collection of discrete carbon fibers. The operating temperature of the fluidized bed is chosen to be adequate to cause the polymer to decompose, leaving clean fibers but not too high that it degrades the carbon fiber substantially [31].

Pyrolysis is a thermal decomposition method for polymers at high temperatures from 350 to 700 °C in the absence of oxygen and an inert atmosphere, e.g., N_2_ [32,33,34]. It allows the recovery of long, high modules fibers, and because of that, it is one of the most widespread recycling processes. Many factors affect the pyrolysis procedure, including the composition of the waste feedstock, reactor type, and process parameters (heating rate, pressure, residence time, temperature) [35,36]. The operating temperatures have a remarkable effect on the fiber’s characteristics (mechanical, electrical, and surface properties) and thus, they need to be adjusted to the type of composites to be treated. A higher temperature can be used, but this will result in some severe degradation of recyclates. Meyer et al. (2009) successfully performed the technical viability of the optimized process parameters using pyrolysis to obtain the reclaimed carbon fibers with properties similar to virgin carbon fibers [37].

Giorgini et al. (2014) used pyrolysis to recover carbon fibers and reported almost total degradation of the polymeric epoxy matrix phase with obtaining solid residue, in which substantially unharmed carbon fibers can be separated, together with oil and gas fractions that can be abused as energy and chemical feedstock [38].

On the other hand, the fluid bed process gives short, recycled carbon fibers in a fluffy form but can treat contaminated wastes with metals. This process produces an excellent fiber product, but it is not in the same form as an existing virgin fiber product.

Although the fluidized bed process is generally more straightforward in theory than pyrolysis, pyrolysis can produce potentially useful organic products from the polymer. There would need to be further processing to isolate them from the mixture of products made, and it seems likely that this would only be cost-effective on an industrial scale. Even though the thermal recycling processes have the advantage of being able to tolerate more contaminated scrap materials, development work is therefore needed to identify how the material can be reprocessed into cost-effective new products. These may have varying degrees of char on the recycled fibers, limiting the reuse options or requiring further processing to remove them.

The dominances of the pyrolysis process are that there is no utilization of chemical solvents and that all the outcomes can be recovered and reused in one form or another. In general, the gases are reused to obtain demanded energy for the process. Recovered carbon fibers have mechanical properties similar to virgin fibers (from 4 to 20% loss in tensile strength) and can be re-manufactured [11,39]. However, the properties of recycled fibers are susceptible to processing parameters. Finally, pyrolysis is the only process used commercially, but the recycled carbon fibers are chopped or milled.

### 3.3. Chemical Recycling

Two principal demands stimulate the progress of chemical recycling technologies, the demand to safely and efficiently process materials challenging to treat with mechanical recycling, and the request to produce high-quality recycled materials [40]. Solvolysis is a chemical treatment using a solvent to degrade the resin. The solvolysis process can recover both the clean fibers and fillers and depolymerized matrix in the form of monomers or petrochemical feedstock [41]. However, it has a low contamination tolerance (e.g., no metals or painting pieces). Solvolysis can offer many possibilities based on catalysts, pressure, temperature, and a wide range of solvents. Solvolysis can be classified according to higher pressure and temperature (temperature > 200 °C) and lower pressure and temperature (temperature < 200 °C), but generally, lower temperatures are requested to degrade the polymer compared to pyrolysis. In chemical recycling, resin degradation is reached using solvents (solvolysis) or water (hydrolysis). The employment of dangerous and concentrated chemicals results in environmental impact, so water or alcohol, which are relatively environmentally friendly, usually replace harmful chemicals [42]. However, it is essential to emphasize that water can be effectively used only in sub- or supercritical conditions.

To achieve a higher dissolution efficiency and faster dissolution rate, catalysts are typically used, and the solvent characteristics are also tuned and optimized. In addition, the dissolution reagents are used to depolymerize the matrix of composites. The reclaimed fiber retains most of its mechanical properties.

Xu et al. (2013) published the tensile strength of the recovered carbon fibers in more than 95% of the virgin ones according to the single fiber tensile test. A mixed solution of hydrogen peroxide and N,N-dimethylformamide was used in this research. According to reported results observed using a scanning electron microscope, the surface of the carbon fibers was smooth with few residues of epoxy resin [43]. Li et al. (2012) used an efficient solution of acetone and hydrogen peroxide for the chemical recycling of carbon fiber/epoxy composites by oxidative degradation. Clean carbon fibers with tensile strength higher than 95% of their original strength can be secured after reacting at 60 °C for 30 min [44].

Sub- and supercritical fluids have also been considered solvents. Semi-long or long recycled carbon fibers can be produced using sub- and supercritical fluids or catalysts such as benzyl alcohol, which dissolves the resin rapidly during the recycling process [45]. However, the utilization of sub- and supercritical fluids can lead to high conditions. Composite recycling by solvolysis has been researched at ambient conditions but at high temperatures and pressures above 300 °C and 50 bar generally for carbon fiber-reinforced composites. Expensive reactors that can withstand high temperatures and pressures and corrosions are requested when sub- and supercritical conditions are used. As a result, there is a trade-off problem between the cost of facilities and the solvent valuables that needs to be solved before applying the solvolysis at an industrial scale.

On the other hand, catalysts are necessary to enhance the reaction at lower temperatures. Nevertheless, this is not always efficient enough to justify the use of catalysts or strong alkaline or acidic conditions that necessarily lead to fiber damages. At ambient conditions, the recycling treatment was performed in extreme conditions, mostly acidic with solutions that can be dangerous in terms of safety and environment.

## 4. Utilization of Sub- and Supercritical Fluids

Sub- and supercritical fluids have distinctive characteristics and valuable potential that may enhance various chemical process operations of multiple materials [46]. The utilization of sub- and supercritical fluids may replace many environmentally harmful solvents currently used in industry, such as organic solvents. Therefore, high-pressure technologies involving sub- and supercritical fluids serve the opportunity to obtain new products with specific properties or to design new processes that are environmentally friendly and sustainable [47,48]. Figure 3 shows a phase diagram of water with marked sub- and supercritical regions [49]. Sub- and supercritical fluids have been reported to achieve efficient and fast resin degradations thanks to enhanced characteristics: liquid-like high mass transport coefficient and pressure-dependent solvent power and gas-like low viscosity and high diffusivity. A clear sub- or supercritical fluid density is easily changed by relatively small variations in pressure and temperature; the viscosity is relatively low but may increase with temperature, whereas the surface tension is essentially nonexistent. Diffusivity is high, which induces interesting transport phenomena in condensed phases [50]. Sub- and supercritical fluids have recently been used in technological areas to develop green processes and can be classed as green reaction media since they are cost-effective and readily available, and they have low potential toxicity [51,52,53]. In addition, they can be recycled afterward by distillation and can dissolve many organic and inorganic compounds. In addition to the dissolving of organic materials, sub- and supercritical fluids can penetrate porous solids, which are still relatively innocuous under atmospheric conditions. Sub- and supercritical fluids are great reaction media for the depolymerization or decomposition of polymers as the reaction is fast and selective. Composite materials such as carbon fiber-reinforced composites can be decomposed into smaller molecular components and fiber materials. Even though it is not necessary, the use of catalysts can remarkably upgrade the decomposition reaction [54,55].

The solvolysis process for carbon fiber-reinforced composites recycling using sub- and supercritical fluids will be reviewed in the following paragraphs. This will manifest the benefits of using this type of solvent regarding its efficiency and reactivity for the decomposition of the polymer matrix and the recovery of the fibers with proper surface and mechanical properties.

Several types of sub- and supercritical fluids (Table 1) have been used for carbon fiber recycling, such as:Water [56,57,58];Methanol [59,60];Ethanol [61];Propanol [62,63,64];Acetone [65].

Under supercritical conditions, water has high diffusivity as well as a high mass transfer coefficient. On the other hand, the viscosity of supercritical water is low, and the dielectric constant is significantly reduced, so the hydrogen bonding essentially disappears. These characteristics allow supercritical water to effectively break down the polymer matrix while causing no damage to fibers [66]. Yuyan et al. (2009) reported that the average tensile strength of the fibers reclaimed using subcritical water was about 98.2% that of the virgin fibers. The scanning electron microscopy and atomic force microscope measurements were employed to observe the fibers’ surface, and no cracks or defects were found. In addition, fibers were clean without resin residues [56].

On the other hand, Piñero-Hernanz et al. (2008) studied the potential for recycling carbon fiber-reinforced epoxy composites in the water at supercritical or near-critical conditions. Experiments were done in a batch-type reactor, and the tensile strength of the reclaimed carbon fibers varied between 90% and 98% compared to the virgin fibers [57]. Additionally, supercritical water was used to solvolysis carbon fiber/thermoset matrix and demonstrate the environmental feasibility of composite recycling [58].

In addition, supercritical methanol was reported to decompose epoxy resin used for matrix resin and recycling of the carbon fiber-reinforced composite. Conditions were 250–350 °C and 100 bar for 5–120 min. When the composite was treated in a semiflow reactor, the recovered carbon fiber had no heat damage and maintained the plain fabric shape [59]. Okajima et al. (2014) recovered carbon fibers with a solid close to virgin fibers using supercritical methanol at 270 °C and 80 bar for 90 min [60].

Hyde et al. (2006) used supercritical propanol to extract and eliminate the epoxy resin from the surface of a carbon fiber composite material. The process was effective at a temperature above 450 °C and pressure above 50 bar. In terms of tensile strength, the recycled fibers were almost as strong as the virgin. It indicated that their structural integrity was minimally damaged [62]. Yan et al. (2016) investigated the effects of degradation temperature on the efficiency of recovering carbon fibers from their epoxy resin composites by supercritical 1-propanol. The obtained results indicated that the mechanical properties of the recycled fibers decreased slightly with temperature [63]. Jiang et al. (2009) researched carbon fiber/epoxy resin composites using supercritical n-propanol and obtained a tensile strength of the recycled carbon fiber close to the corresponding as-received carbon fibers [64].

Okajima and Sako (2019) investigated the chemical recycling of carbon fiber-reinforced composites using supercritical acetone. They reported that the decomposition efficiency increased with increasing reaction pressure and acetone density, to a maximum value of 95.6% at 350 °C, 140 bar, and 60 min [65]. Furthermore, Okajima et al. (2019) used subcritical acetone and various supercritical solvents (methanol, ethanol, 1-propanol, 1-butanol, 2-butanol, tert-butanol, acetone, and methyl ethyl ketone) as solvents in a composite recycling procedure at 320 °C in the reaction time range of 6–120 min. The decomposition rate depended on the solvent, but sub- and supercritical acetone were selected as optimal for fast degradation of the matrix resin. The recovered carbon fibers saved the shape of the fabric sheets, and their tensile strength reduction was insignificant [67].

Sokoli et al. (2017) compared the degradation of composites using near-critical water and supercritical acetone and varied parameters such as temperature (in a range from 260 to 300 °C), pressure (from 60 to 300 bar), and composite/solvent ratio (from 0.29 to 2.1 g/mL). They determined the exact conditions for achieving nearly complete degradation of the resin using supercritical acetone. The tensile strength of recovered carbon fibers was retained using solvents, water, and acetone [68].

**Table 1 materials-14-04191-t001:** Most used sub- and supercritical fluids as a solvent in carbon fiber-reinforced composite recycling.

Fluid	Critical Data *	Reference
water	Tc = 647.1 K; Pc = 220.6 bar	[69]
methanol	Tc = 512.6 K; Pc = 81.0 bar	[70]
ethanol	Tc = 514.0 K; Pc = 61.4 bar	[71]
propanol	Tc = 536.8 K; Pc = 52.0 bar	[71]
acetone	Tc = 508.0 K; Pc = 48.0 bar	[72]

* Pc—critical pressure; Tc—critical temperature.

Chemical recycling with sub- and supercritical fluids is a more recent approach; nevertheless, it is already recognized for producing recycled carbon fibers with virtually no mechanical degradation, especially when using sub- and supercritical alcohols and enabling recovering valuable chemicals from the matrix [73]. They are inexpensive and easy to handle, exhibiting low critical pressures (typically 20–60 atm) but high critical temperatures (usually 200–300 °C). However, the extraction process using sub- and supercritical alcohols does not significantly alter the thermal stability of the carbon fibers. Piñero-Hernanz et al. (2008) obtained carbon fibers that retain 85–99% of the strength of the virgin fibers using batch and semi-continuous-type reactors. Subcritical and supercritical alcohols (methanol, ethanol, 1-propanol, and acetone) were employed as reactive-extraction media in their study [61].

Generally, recycling using sub- and supercritical fluids is a quick and simple method and can be operated semi-continuously [74]. Furthermore, the decomposition of polymers proceeds rapidly and selectively, and the fibers compare favorably to the virgin fibers, with only a slight loss of tensile strength [75,76]. In addition, compared to thermal treatments, preservation of brittleness, length, and orientation is better [77]. Total removal and subsequent recovery of the epoxy resin residue are possible. Resin is immediately removed by dissolution and can be recovered ex situ from the solution by evaporation. The quality of the recovered fibers in terms of surface texture is high; the fibers appear to be utterly free from epoxy resin polymer and have a clean and smooth surface [78]. Furthermore, there is no apparent reduction in diameter or surface scratches on the fibers (the extraction process does not appear to damage the fibers).

## 5. Table Review

Recycling methods and required conditions for their realization, such as temperature, pressure, time, heating rate, atmosphere, and solvent, and obtained tensile strength of recovered fibers compared to virgin fibers, have been reviewed in Table 2.

## 6. Conclusions and Future Perspectives

Up-to-date changes to waste management legislation and possible future directions mean that recycling directions need to be in place. The composite industry is under intense pressure to provide viable recycling scenarios for their materials if they want to continue to have a place in the market. A review of different existing recycling technologies for carbon fiber-reinforced composites has been presented in this paper. For each recycling technology, specific advantages and drawbacks have been given.

Chemical recycling gives the recovery of long fibers with good mechanical properties in regard to mechanical recycling. Accordingly, fibers resulting from this recycling technique can be reused in the re-manufacturing of new composites. The loop in the carbon fiber-reinforced composite life cycle can then be closed properly.

However, chemical processes may employ solvents that can have adverse effects on the environment and human health. Thus, to reduce these impacts, sub- and supercritical fluids have been used as green reaction media to solve carbon fiber-reinforced composites. Compared to traditional recycling techniques, solvolysis is an excellent alternative to recover fibers. It provides high mechanical properties and fiber length and a high potential for material recovery from resin. Even though this process has shown excellent results for recycling carbon fiber-reinforced composites at the laboratory scale, further investigations need to be managed to design a cost-effective pilot plant to be used at an industrial scale.

Further research studies have been undertaken as this material has been used more frequently in new applications. Research and developments on emerging green technologies using sub- and supercritical fluids are anticipated to expand soon. Overall, the recycling of carbon fiber-reinforced composites using solvolysis in sub- and supercritical conditions has been proven to be a great alternative, and it has been highlighted in this review.

The topic that is discussed is innovative and sustainable. The major limitation on the way to its (possible) industrial application is the scaling-up. This challenge is related to the knowledge of the basic process properties such as thermodynamic and mass transfer data. This should be determined for each system experimentally at the conditions pertaining to the process condition. Therefore, this process is challenging and requires fundamental research of thermodynamic and mass transfer data.

## Figures and Tables

**Figure 1 materials-14-04191-f001:**
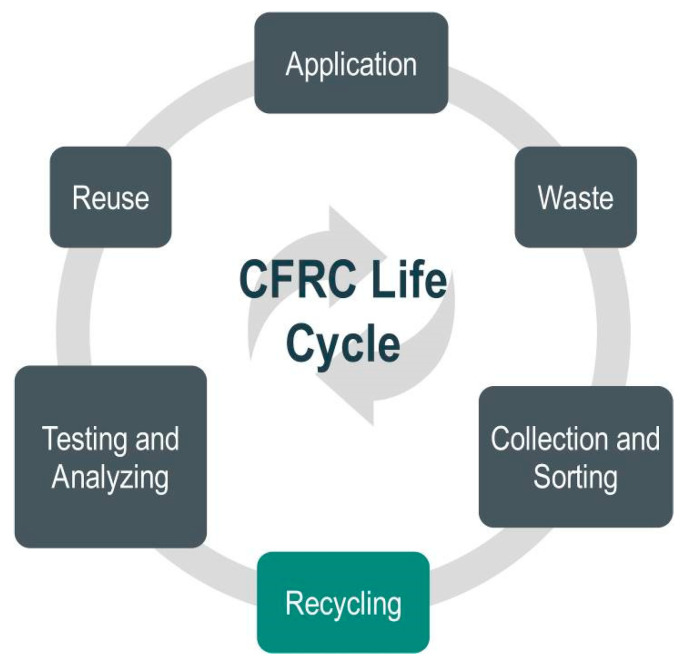
A scheme of carbon fiber-reinforced composite (CFRC) life cycle.

**Figure 2 materials-14-04191-f002:**
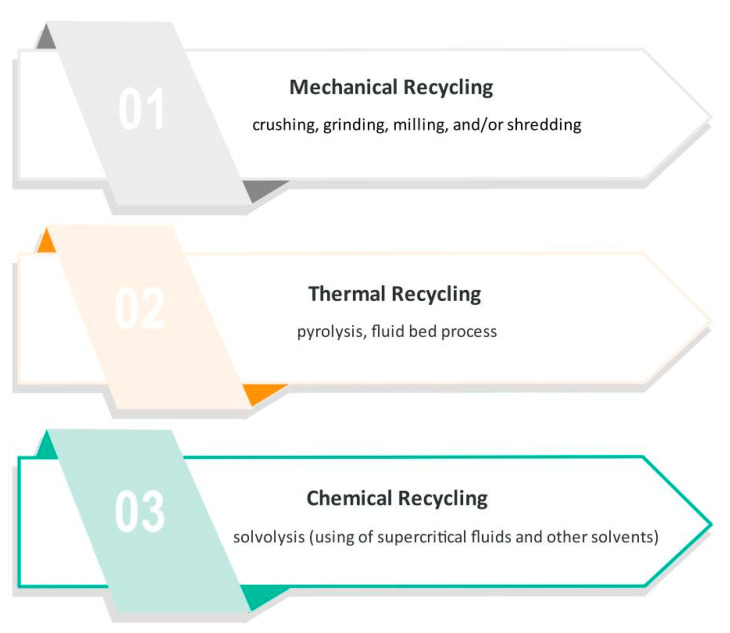
The most usual procedures for recycling carbon fiber-reinforced composites.

**Figure 3 materials-14-04191-f003:**
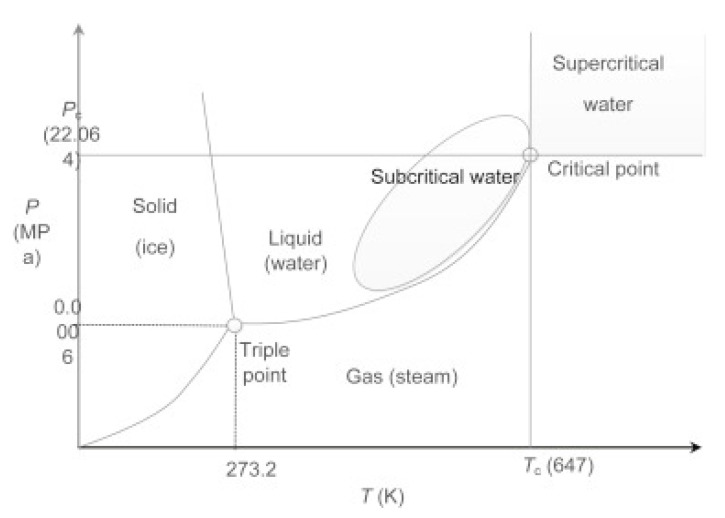
Phase diagram of water with marked sub- and supercritical region, where Pc represents critical pressure and T_C_ critical temperature. Reprinted with permission from ref. [49]. Copyright 2016 Elsevier.

**Table 2 materials-14-04191-t002:** Literature summary on most used composite recycling methods.

Recycling Method	Parameters	Tensile Strength of Recovered Fiber	Reference
Thermal (pyrolysis)	400–600 °C	undefined	[37]
	10 °C/min		
	120 min		
	air and N_2_ atmosphere		
Thermal (pyrolysis)	500–600 °C	undefined	[38]
	150 min		
Chemical (solvolysis)	90 °C	τ > 95%	[43]
	30 min		
	in a solution of H_2_O_2_/DMF (1:1, *v*/*v*)		
Chemical	60 °C	τ > 95%	[44]
	30 min		
	Ac and H_2_O_2_		
Chemical (subcritical)	260–290 °C	τ = 98.2%	[56]
	10–400 bar		
	75–105 min		
	H_2_O		
Chemical (near- and supercritical)	250–400 °C	90% < τ < 98%	[57]
	40–270 bar		
	30 min		
	H_2_O		
Chemical (supercritical)	400 °C	undefined	[58]
	250 bar		
	30 min		
	H_2_O		
Chemical (supercritical)	270 °C	τ = 91%	[60]
	80 bar		
	90 min		
	MeOH		
Chemical (supercritical)	above 450 °C	τ ≈ 95.4%	[62]
	above 50 bar		
	several minutes		
	PrOH		
Chemical (supercritical)	260–340 °C	94.6% < τ < 95.2%	[63]
	120 min		
	1-PrOH		
Chemical (supercritical)	310 °C	88.6% < τ < 99.1%	[64]
	52 bar		
	20 min		
	n-PrOH		
Chemical (supercritical)	PrOH	undefined	[65]
	20–140 bar		
	60 min		
	Ac		
Chemical (sub- and supercritical)	320 °C	negligible reduction	[67]
	10 bar		
	20 min		
	MeOH, EtOH, 1-PrOH, 1-BuOH, 2-BuOH, tert-BuOH, Ac, MEK		
Chemical (sub- and supercritical)	300–450 °C	85% < τ < 99%	[61]
	47–153 bar 15.5 min		
	MeOH, EtOH, 1-PrOH, Ac		

## Data Availability

Data sharing is not applicable.

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
