# Peer review of "Recycling of Carbon Fiber-Reinforced Composites—Difficulties and Future Perspectives"

_materials, 2021, doi:10.3390/ma14154191_

Round 1

Reviewer 1 Report

The manuscript successfully identified and aptly described different methodologies to recycle carbon fiber reinforced composites. As the authors truly stated, a review of this kind is not adequately available. Thus, I support publication of this manuscript with minor revisions. 

I believe, if the authors address the following issues, this manuscript will be more attractive to the readers.

  1. In the introduction, please include and elaborate typical applications of carbon fiber reinforced composites.
  2. Line 79, "To ease the continued use .... some applications ...", please give some examples of such applications.
  3. Line 149, "On the other hand, fluid bed process ...", it would be easier for readers to connect if you can briefly describe the "fluid bed process".
  4. Line 164, "Loss in tensile strength ...", please provide appropriate citation for such claims.
  5. In the Table 1, authors used abbreviated form of solvent names but can be confusing to readers who are not familiar with such abbreviations. 
  6. Finally, I found two typos needed to fixed. Line 134, nitrogen is N2, not N2. Line 350, starts with "Researchers and developments ..." which should be "Research and developments ..."

Reviewer 2 Report

In this review authors summarized the three processes available for recycling CF in composites, with special emphasis in chemical recycling. The paper is evidence-based and well designed with data presented in Table 2 allowing the reader to get an overview of the results of different studies. However, in my opinion, review articles need to go beyond mere description and ‘state-of-the-literature’ summaries and authors should also offer new insight into understanding by addressing new questions and stimulate further empirical work. So, in my opinion authors should add:

1) Lines 176-183: i) refer that solvolysis can be classify based on higher temperature and pressure (HTP) with temperature > 200 °C and lower temperature and pressure (LTP) with temperature < 200 °C.  ii) Eventually refer that in Chemical recycling resin degradation is either achieved using solvents (solvolysis) or water (hydrolysis), and the use of hazardous and concentrated chemicals results in environmental impact, so the harmful chemicals are replaced with water and alcohol at supercritical conditions. Is important to understand that only in that conditions water can be effectively used.

2) contribute with new innovative questions and introduce some recommendations for future works that will allow move beyond the current knowledge. The point is how can we make the scale-up from laboratory-scale solvolysis at supercritical conditions into a fully functional commercial scale.

Reviewer 3 Report

The manuscript entitled "Recycling of Carbon Fiber Reinforced Composites – Difficulties and Future Perspectives" is a well-written review paper relating to the problem of recycling materials of complex and problematic composition. The article reads quickly and pleasantly, and the content is sufficient to highlight the issues and summarize the global efforts on the issue. The authors also, very importantly, have analyzed the trends they observe from their own experience and literature review. Also, the simplicity, aesthetics, and readability of the posted images are conducive to understanding the content being read. In my opinion, the article is suitable for publication as submitted. 
